# Counseling and Prescription of Physical Exercise in Medical Consultations in Portugal: The Clinician’s Perspective

**DOI:** 10.3390/healthcare13090986

**Published:** 2025-04-24

**Authors:** Rita Quintas Oliveira, Edite Teixeira-Lemos, Jorge Oliveira, Joana Morais, Diogo Miguel, Luís Pedro Lemos, João Páscoa Pinheiro

**Affiliations:** 1Faculty of Medicine, University of Coimbra, 3000-548 Coimbra, Portugal; reabmedica@hotmail.com; 2CERNAS-IPV Research Centre, Polytechnic University of Viseu, Campus Politécnico, 3504-510 Viseu, Portugal; joliveira@esav.ipv.pt; 3Trás-os-Montes e Alto Douro Health Local Unit, 5000-508 Vila Real, Portugal; joanaquintasm@hotmail.com; 4Viseu Dão-Lafões Health Local Unit, 3504-509 Viseu, Portugal; diogo.miguel.6619@ulsvdl.min-saude.pt; 5Nuclear Medicine Department, Hospitais da Universidade de Coimbra, Unidade Local de Saúde de Coimbra, 3000-548 Coimbra, Portugal; 11652@ulscoimbra.min-saude.pt

**Keywords:** exercise is medicine, counseling, prescription, enabling factors, hindering factors

## Abstract

**Background/Objectives**: Physical exercise (PE) is essential in promoting health and quality of life and protecting against chronic diseases. Health professionals are identified as key figures in promoting and prescribing PE, yet various factors may impact this during consultations. This study aims to assess Portuguese specialist physicians’ understanding of the importance of PE prescriptions. It will also investigate the approaches they utilize in promoting and prescribing PE, their knowledge of incorporating this practice into their consultations, and the major facilitators or barriers to prescription. **Methods**: A cross-sectional observational study was conducted using a validated questionnaire distributed via email by the Centre Regional Section of the Portuguese Medical Association to physicians. The data were analyzed using descriptive and inferential statistics. **Results**: In total, 414 responses were collected, with participants representing different medical specialties. The participants were primarily women (62.8%) with a mean age of 49.9 ± 14.9 years. While 85.5% of physicians promoted and prescribed PE, recognizing its cardiovascular and metabolic health benefits, only 24.0% received specific training, and 73.7% were unaware of relevant guidelines. Older male physicians (over 60 years old) expressed more confidence in PE prescriptions, while younger ones were more familiar with electronic prescribing tools. Identified barriers included patient compliance (42.3%), a lack of accessible PE resources (18.4%), and limited consultation time (17.4%). Most physicians (64.4%) relied on oral counseling for prescriptions. **Conclusions**: Most Portuguese specialist doctors recognize the benefits of PE prescriptions. However, barriers like inadequate training and patient compliance hinder PE implementation. Enhanced training and resources are vital for effectively integrating PE into clinical practice.

## 1. Introduction

Physical activity (PA) is defined by the World Health Organization (WHO) as “any bodily movement produced by skeletal muscles that requires energy expenditure” [1]. It entails daily activities such as walking, climbing stairs, doing household chores, and physical exercise (PE). The latter is “planned, structured, repetitive, intentional and aims to improve or maintain one or more components of physical fitness” [1,2,3].

Therefore, PE plays an important role in enhancing health and quality of life [4,5], enabling the maintenance of physical and mental well-being [1,6,7,8,9,10], and controlling symptoms associated with psychiatric illnesses [11] such as depressive and anxiety disorders [12,13,14] and dementia [1,14,15]. It is also a protective factor against cardiovascular [1,6,7,11,12,16,17], metabolic [1,6,11,12,17], neoplastic [11,12], and musculoskeletal [14,16] diseases, reducing the prevalence of these and other chronic noncommunicable diseases [7,12,14,18].

A lack of physical activity is currently a major public health problem and one of the main risk factors for mortality [3,6]. It leads to the development of a range of pathologies [19,20] that will compromise quality of life. Portugal has a significant sedentary lifestyle among young people and adults [3]. Data from the National Food, Nutrition, and Physical Activity Survey reveal that a significant portion of Portuguese adults fall into the categories of either moderate activity (30.6%) or low activity (42.3%) [21]. While physical activity levels seem to remain steady, recent findings from motion sensors show that only around 15% of youth, 71% of adults, and 31% of older adults in Portugal are meeting the recommended guidelines for physical activity [22]. This behavior is a public health concern. A recent economic assessment warns that if physical inactivity persists, we could see nearly 500 million new cases of avoidable major noncommunicable diseases in the coming decade, leading to projected healthcare expenses of about USD 47.6 billion annually [23].

In this context, it is important to mention the European Union (EU) Guidelines for PA [19] and the National Strategy for the Promotion of PA, Health and Well-being (2016–2025) [20], resulting from the European Strategy for PA (2016–2025) [10], developed by the European Commission in collaboration with the WHO European Committee and the Directorate-General for Health (DGS). The WHO has also developed a Global Action Plan for PA (2018–2030) [1]. These documents aim to make recommendations for policies, strategies, and specific objectives for promoting PA and advocate a systemic approach that encourages the creation of healthy lifestyles.

Several guidelines [1,3,10,20,24] have been created to reinforce the importance and need to implement measures in multisectoral systems, managed by policies to promote the prescription of PE by doctors [3,17], persistent monitoring, and individual and social motivation to increase the practice of PE.

In Portugal, following these guidelines, among other measures, a tool for prescribing PE was implemented in December 2017 on the Electronic Medical Prescription digital platform. This tool allows doctors to prescribe personalized guidelines based on the patient’s activity level and motivation. This tool makes it possible to apply strategies to initiate, maintain, or consolidate the promotion of PE practice [8]. It is also accessible to other health professionals via the DGS website, promoting a broader approach to promoting PE.

Doctors are key players in public health promotion [8] and credible sources for the population [16,25,26]. By regularly promoting and prescribing PE in their consultations, they ensure the creation of encouraging conditions for a more active and healthy lifestyle [1,6], faster recovery from certain illnesses, and delaying their progression [1].

Counseling for the promotion of PE is understood as an intervention through oral encouragement, instructions, and verbal or written recommendations endorsed by health professionals [3,25,26].

Prescribing PE is a medical act that should begin with a detailed assessment of the patient’s state of health, physical and functional fitness, and body composition, followed by clarification on how to perform certain physical exercises, as well as the precautions to be taken [3]. It is extremely important to establish constant monitoring in this process, which allows for greater agility in changing aims, if necessary [3]. Regardless of the specificities, in studies carried out in Portugal [25], Germany [16], Saudi Arabia [18,27], Canada [28], Thailand [4], and Brazil [29], it is clear that there are several factors influencing the prescription of PE, namely, the medical specialty [17], the lack of specific knowledge [4,9,17,18], the PE practiced by the doctors themselves [9,17], the lack of time for consultations [4,9,16,17,18,28,29], and the lack of available resources [14,28,29]. The relevance of the health status assessment, the patients’ motivation, and their willingness to achieve the predetermined goals also influence this prescription [9,16,28].

The aim of this study is to assess the understanding of specialist doctors, registered with the Portuguese Medical Association, regarding the counseling and prescription of PE in medical consultations, considering (1) the perceived importance of prescribing PE; (2) the promotion and skill for the incorporation of PE prescription; and (3) the main facilitating and constraining factors that influence the implementation of this prescription as a regular practice by professionals. By addressing these objectives, this research aims to identify gaps in knowledge and practice among specialist doctors, especially in the context of the increasing sedentary lifestyle in Portugal. Additionally, it strives to provide valuable insights that can contribute to developing targeted training programs and guidelines that enhance the routine integration of PE into clinical practice, aligning with national and international health promotion strategies.

## 2. Materials and Methods

### 2.1. Study Design

This cross-sectional observational study is based on a questionnaire applied to specialist physicians of all specialties registered with the Portuguese Medical Association. It is important to note that in Portugal, exercise prescriptions are viewed similarly to any other prescription. Consequently, only licensed physicians can provide customized, written prescriptions for physical exercise as part of treatment plans. The act of prescribing exercise is a medical procedure that may only be performed by authorized medical doctors. This study was approved by the Faculty of Medicine Ethics Committee of the University of Coimbra (CE-074/2024) on 16 May 2024.

The exclusion criteria defined were not completing the questionnaire in full and having completed their Bachelor’s/Master’s degree after 2018, which does not allow, from a chronological point of view, the completion of any specific training internship until January 2025. Participation in the study was voluntary and anonymous and was subject to the acceptance of the Informed Consent form on the first page of the questionnaire. Data collection took place from 15 December 2024 until 31 January 2025.

The diffusion of a questionnaire requires its preparation and validation. These procedures are described below.

### 2.2. Minimum Sample Required

The study population comprises specialist doctors of all specialties registered with the Portuguese Medical Association.

As a first step, and to ensure that the results obtained are representative and robust, the minimum necessary sample was calculated beforehand using the Cochran Formula (1) [30]. We knew that 40,716 specialist doctors were registered with the Portuguese Medical Association in December 2023 [31] and took into account the values of the assumptions (defined in the description of the parameters relating to Formula (1)):(1)$$n=\frac{Z^{2}\times p\times \left(1\;-\;p\right)\times N}{Z^{2}\times \left(N\;-\;1\right)+Z^{2}\times p\times (1\;-\;p)}$$
where n = sample size; Z = value of the normal distribution corresponding to the 95% confidence level (1.96); *p* = unknown expected proportion of the population (0.5 for maximum variability); E = 5% margin of error (0.05); and N = population size (40,716). Therefore, the estimated minimum sample size was 381.

### 2.3. Questionnaire Development

Planning, design, preparation, and validation were carried out to create the questionnaire; these were essential steps prior to its dissemination and made it a more robust instrument [30,31]. Hence, in the first phase, a pre-questionnaire was created, divided into four sections, with closed-answer questions, considering (1) personal and professional data, (2) the importance of PE in healthcare and health status, (3) the promotion/behavior and competence of each doctor in prescribing PE, and (4) prescribing procedures and their limitations.

In addition to the main questions in the pre-questionnaire, each one was assessed for its relevance to the final questionnaire, according to the study’s objectives, using a Likert scale (1–5), where one corresponds to ‘Not at all relevant’ and 5 to ‘Extremely relevant’. A ‘Comments’ field made it possible to include suggestions for changing or improving any questions.

This pre-questionnaire was uploaded to the Microsoft Forms platform, with the first page being the Informed Consent form.

This pre-questionnaire was evaluated using a ‘snowball’ sampling process to include specialist doctors who voluntarily agreed to participate and passed it on to other people they knew. The pre-questionnaire was sent out by e-mail, and 15 specialists responded.

Data obtained from the pre-questionnaire were analyzed, leading to the inclusion of new items such as ‘Main place of work (district)’ and ‘Does not apply to usual clinical practice’ in the question, ‘What is the main reason for not prescribing physical exercise?’.

The relevance of the pre-questionnaire questions was assessed using Cronbach’s alpha coefficient, which was 0.969, indicating excellent internal consistency between the items. Additionally, Cronbach’s alpha coefficients of 0.749 and 0.758, respectively, for the multinomial (Likert scale) and dichotomous (yes/no) variables confirmed the internal consistency of both the pre-questionnaire and the definitive questionnaire. This suggests that the questions effectively assess the intended objectives related to the prescription of PE.

Finally, after these procedures, the definitive questionnaire was established.

### 2.4. Application of the Questionnaires

The Centre Regional Section of the Portuguese Medical Association distributed the final questionnaire via e-mail using the Microsoft Forms platform, which allowed it to cover the whole of Portugal.

### 2.5. Statistical Analysis

The reliability of the pre- and final questionnaires was tested by assessing the internal consistency of the blocks of categorical and binominal questions of Cronbach’s alpha. At the same time, the relevance of the pre-questionnaire questions was checked by analyzing frequencies and verifying their internal consistency using Cronbach’s alpha.

The data collection subsequently made it possible to characterize the sociodemographic, academic, and professional profile of the sample, considering the dichotomous variable gender, male or female; the continuous variable age, which was coded into the following classes: [29–40], ]40–60], and ]60–83]; the continuous variable year of completion of Bachelor’s/Master’s degree, which was also coded into classes: [1967–1975], ]1975–1985], ]1985–1995], ]1995–2005], ]2005–2018]; the training institution (Faculty); the main place of work (District); and medical specialty.

The answers to the final questionnaire were analyzed using descriptive and inferential statistics. The Kolmogorov–Smirnov test was carried out beforehand to assess the normality of the data distribution of all the variables under study. Depending on the results, parametric or non-parametric tests were carried out.

The representativeness of the responses, by medical specialty, was ascertained using Pearson’s correlation and the paired samples Student’s *t*-test, considering the proportions of specialist doctors registered with the Portuguese Medical Association and of responses to the questionnaire, by specialty.

To assess the influence of age on whether or not PE was promoted/prescribed, the independent samples Student’s *t*-test was used. The influence of gender on this variable was analyzed using Fischer’s test (χ^2^).

The influence of the independent variables gender and age classes on the dependent variables habit of prescribing PE, level of competence/difficulties in prescribing PE, and main procedure in prescribing PE was assessed using tests from χ^2^ and the analysis of the adjusted residuals.

Using the χ^2^ test and the analysis of adjusted residuals, the main challenge for promoting and prescribing PE and the reason for not promoting and prescribing PE were analyzed.

All statistical analyses were conducted using the Statistical Package for Social Sciences software, v.29, with a 95% confidence level (α = 0.05).

## 3. Results

### 3.1. Sociodemographic, Academic, and Professional Characterization of the Sample

Out of 427 questionnaires completed on the Microsoft Forms platform, 13 were removed according to the exclusion criteria.

The final sample comprised 414 specialist doctors, with an average age of 49.9 ± 14.9 years old, ranging from 29 to 83 years. Among participants, the average age for male doctors was 53.6 ± 16.6 years old, while the average age for female doctors was 47.8 ± 13.4 years old. Of the total sample, the majority were female (n = 260; 62.8%). Most specialist doctors who answered the questionnaire graduated from the Faculty of Medicine of the University of Coimbra (n = 227; 55.2%) and completed their Bachelor’s/Master’s degree between 2005 and 2018 inclusive (n = 193; 47.0%). Table 1 summarizes the respondents’ sociodemographic, academic, and professional characteristics.

Regarding medical specialty, no response was received for 8 of the 50 medical specialties (Cardiac Surgery, Maxillofacial Surgery, Thoracic Surgery, Clinical Pharmacology, Medical Genetics, Tropical Medicine, Neurosurgery, and Neuroradiology), whereas the most representative were General and Family Medicine (24.9%), Internal Medicine (14.0%), and Pediatrics (8.2%). However, there was a very high correlation (R = 0.964; *p* < 0.001) between the number of respondents, by specialty, and the number of specialist doctors registered with the Portuguese Medical Association in 2023, by specialty (Figure 1). Cumulatively, using Student’s *t*-test for paired samples, we found no significant differences (*p* = 0.996) in the proportion of responses, by specialty, between the number of responses obtained and the specialist doctors registered with the Portuguese Medical Association.

### 3.2. Factors Influencing the Promotion/Prescription of PE

Of the respondents, 354 (85.5%) specialist doctors promote and prescribe PE.

Fischer’s exact test (χ^2^) showed that the gender of medical specialists does not influence the promotion and prescription of PE (*p* = 0.195).

On the other hand, Student’s *t*-test showed that the age of doctors influences (*p* < 0.001) the promotion and prescription of PE. The results show that the highest percentage of non-prescribers (66.7%) is concentrated in the younger medical class ([29–40] years) and that the highest concentration of prescribers (64.3%) is above the age of 40.

### 3.3. Specialist Doctors’ Perception of the Importance of PE in Healthcare and Health Status

Figure 2 shows that the majority of doctors agree with the statements presented, with a greater uniformity of responses regarding the statement that PE contributes to improving cardiovascular health and metabolic condition (90.6% totally agree).

### 3.4. Intentions/Habits in Prescribing PE, Level of Competence/Difficulties in This Practice, and Main Procedures Adopted

A significant commitment to promoting and prescribing PE was evident among the respondents. Overall, 85.5% of the physicians indicated that they actively promote and prescribe PE. In Table 2, we delineate how both age and gender factors influence specific intentions and habits in prescribing PE.

Gender does not influence the habit of prescribing exercise or the future intentions of specialist doctors concerning this practice (*p* = 0.344); however, the data highlight that male physicians reported more training specific to PE prescription than their female counterparts (30.7% vs. 20.3%; *p* = 0.037). This distinction might underscore a need for increased access to training resources for female physicians as they engage with PE in clinical practice.

The data indicates that age significantly affects the intention to prescribe PE (*p* < 0.001). Most doctors who promote and prescribe PE (82.2%) have been doing so for more than 6 months, and this is more evident in doctors aged 40 and over. Conversely, 18.7% of doctors in the [29–40] age group express a strong intention to begin promoting and prescribing PE in the near future. This suggests a burgeoning interest in younger physicians, potentially reflecting their perspectives on the evolving role of exercise in healthcare.

Table 2 also illustrates age-related confidence levels in various aspects of PE prescription. Confidence levels varied considerably, as male doctors reported significantly higher confidence in assessing patient needs and prescribing PE than female doctors (71.7% vs. 48.9%; *p* < 0.001). Medical specialists in the [60–83] age group showed the highest confidence in assessing patient needs (79.3%), discussing PE recommendations (86.2%), and monitoring/adapting PE prescriptions (57.8%), supported by statistical significance (*p* < 0.001). Male doctors similarly exhibited greater confidence in these aspects, with 87.4% feeling adept at discussions and 57.5% in monitoring (both *p* < 0.001).

These findings suggest that experience may be crucial in enhancing confidence and competence in PE prescription.

Notably, neither gender nor age influenced the knowledge of PE prescribing guidelines (*p* ≥ 0.05), with an alarming 73.7% of respondents lacking familiarity with these guidelines (n = 261). This suggests a pressing need for enhanced education on PE guidelines in medical training and continued professional development.

On the other hand, doctors in the [29–40] age group were more aware of the functionality associated with the Electronic Medical Prescription PE prescription tool (52.0%; *p* < 0.001), and gender did not influence this knowledge (*p* = 0.365). This suggests that younger physicians may be more adept at utilizing digital resources effectively within their practice.

When it comes to the main procedure adopted by specialist doctors, most (64.4%) resort to oral counseling for autonomous or supervised PE practice in their consultations. This reliance on verbal recommendations reflects a common approach among healthcare providers; however, it also highlights a need for structured, written guidance that could reinforce patient understanding and adherence.

The preferred methods for prescribing PE showed no significant age influence (*p* = 0.142), but gender impacted the type of strategies employed. Male doctors were more likely to prescribe detailed exercise plans than female doctors (71.4%; *p* = 0.032), indicating potential gender-based differences in approaches to PE prescription.

### 3.5. Relationship Between the Promotion/Prescription or Non-Promotion/Prescription of PE and the Respective Challenges/Motives

The distribution of challenges in prescribing or reasons for not prescribing among doctors differs according to whether they prescribe or not (*p* < 0.001). It is clear that non-application to clinical practice (48.3%) is the main reason for not prescribing PE (Table 3). For those who do prescribe PE, the challenge of patient adherence (48.6%) emphasizes a critical hurdle in achieving effective health outcomes.

## 4. Discussion

The practice of regular PA and the maintenance of adequate levels of PE is a widely recognized and valued subject in the field of prevention. Several authors have emphasized its crucial role as a non-pharmacological therapeutic alternative for several pathologies [4,5,7,32]. The DGS has already recognized the importance of both PA and PE for patients with chronic pathologies, implementing initiatives to promote them [13]. However, there are still many obstacles to overcome, including the lack of adequate training [4,9,15,33].

The number of specialist physicians from all over the country, registered with the Centre Regional Section of the Portuguese Medical Association, who contributed to this study (414) was higher than the estimated minimum sample required (381). The response to this online questionnaire shows the interest shown in the topic and its relevance and indicates that the results analyzed can be inferred for the population, making the data analysis more robust.

The majority of the sample (62.8%) is made up of women, reflecting a growing trend towards the feminization of the medical profession in Portugal [34]. The very high correlation (R = 0.964; *p* < 0.001) between the number of respondents by medical specialty and the number of specialist doctors registered with the Portuguese Medical Association suggests that the sample is representative of the actual distribution of specialist doctors in Portugal by medical specialty, allowing for more robust inferences about the prescription of PE among specialist doctors. In fact, the data obtained in this study result from different professional experiences, suggesting that the prescription of PE is considered relevant in several specialties.

### 4.1. Specialist Physicians’ Understanding of PE Prescription Importance

The results show that medical specialists highly recognize the health benefits of PE. There was greater uniformity of responses regarding the positive impact of PE on cardiovascular health and metabolic status. However, it is important to note that doctors of certain specialties, such as Cardiac Surgery, Maxillofacial Surgery, Thoracic Surgery, Clinical Pharmacology, Medical Genetics, Tropical Medicine, Neurosurgery, and Neuroradiology, did not provide responses. This lack of participation may be attributed to the limited direct patient interaction typical of these fields. Physicians in these specialties often do not engage in clinical practices where prescribing PE is within their scope of expertise, which may have influenced their decision not to participate. In fact, 85.5% of respondents promote and prescribe PE, a number very similar to data collected in previous studies conducted in Portugal (84.6%) [9], Brazil (81.2%) [29], and Ireland (86.4%) [15]. This high level of agreement emphasizes the importance of training doctors so that they can translate this knowledge into concrete actions, promoting the adoption of PE prescription as an essential part of clinical practice. This level of agreement underscores the necessity for comprehensive training programs for physicians, enabling them to effectively translate their knowledge into practical applications. By doing so, we can foster the integration of PE prescription as a fundamental component of clinical practice, ultimately enhancing patient health outcomes.

### 4.2. Physicians’ Skills for Integrating PE Prescription into Their Consultations

This study demonstrates that while a significant majority of physicians recognize the importance of promoting and prescribing PE, their confidence in their skills to implement this varies markedly.

Notably, only 24.0% of those who prescribe PE have received specific training in this area, revealing a critical gap that may affect their competence and confidence when discussing and prescribing exercise regimens to patients.

Another relevant aspect is the lack of familiarity with the guidelines for PE prescription, stated by 73.7% of physicians, suggesting that the majority may not feel adequately prepared to integrate this practice effectively into their consultations. These data align with a study in Saudi Arabia [27] in which only a small percentage of doctors (5.4%) were knowledgeable about relevant guidelines. These results reinforce the need for greater diffusion of specific guidelines for PE prescription, including them into medical curriculum and continuing training [8,27]. Currently, at the Faculty of Medicine of the University of Coimbra, the Curricular Unit on PE prescription remains elective despite high student demand, indicating a discrepancy between interest and availability of formalized training. This situation calls for a structural change in medical curricula to ensure that PE receives appropriate emphasis, equipping future physicians with the knowledge and skills necessary for effective exercise prescription. In addition, our findings indicate that doctors’ confidence in their ability to assess patient needs for prescribing PE, to discuss PE recommendations with patients, and to monitor and adjust PE prescriptions was higher among older doctors (*p* < 0.05), suggesting that experience and continuous training play a vital role in strengthening the skills needed to integrate PE prescription into clinical practice. More experienced physicians may have had greater opportunities for practical application and mentorship regarding PE prescription compared to their younger counterparts.

Conversely, younger physicians showed higher familiarity with Electronic Medical Prescription tools (52.0%), suggesting that they may be more adept at leveraging digital resources and clinical decision-support systems to enhance patient care. This points to a potential integration of modern technology and decision-support systems into the clinical workflow, which could facilitate better exercise prescription practices.

To maximize the efficacy of PE as a therapeutic intervention, it is imperative that medical education programs address these training gaps. Implementing structured curricula and ongoing training focused on PE guidelines will boost physicians’ confidence and empower them to prescribe exercise as a vital component of patient health. Additionally, fostering a collaborative approach within healthcare that involves exercise physiologists and other professionals can improve the support available to physicians, ultimately enhancing patient outcomes.

### 4.3. Procedures Adopted in the Promotion and Prescription of PE

The procedures adopted by specialist doctors when promoting and prescribing PE reflect a reliance on alternative approaches. Many physicians prioritize general recommendations and motivational verbal counseling during consultations, rather than establishing formal, structured PE prescription schemes. Additionally, many specialists, influenced by their medical training, often prefer to prescribe medications or consider alternative treatment options, such as surgery, rather than viewing PE as a viable treatment strategy. This reliance on less formal methods has been similarly noted by O’Brien et al. [15]. At the same time, although most doctors showed confidence in their abilities to prescribe PE, many said that they did not know the guidelines or new tools to promote this practice, suggesting that the prescription of PE may not be adequately implemented by doctors [27,29,35].

### 4.4. Identification of Factors That Facilitate or Hinder the Prescription of PE

Among specialist doctors who do not prescribe PE, the main barrier identified was the perception that this practice does not apply to their clinical reality (48.3%; *p* < 0.001). This may reflect diverse causes, including the perception that PE does not fit the needs of their patients, does not fall within their specialty, or even that a structured model for its prescription is absent from their usual clinical practice [2].

On the other hand, for doctors who prescribe PE, the biggest challenge encountered in this practice was the lack of patient compliance (48.6%; *p* < 0.001). This is also evidenced in studies carried out in Saudi Arabia [18,27], Canada [33], Ireland [15], Mexico [35], and Portugal [9]. These results suggest that, despite doctors’ intention to promote and prescribe PE, the challenges related to patient motivation and adherence to medical recommendations remain a central issue in clinical practice [15]. The inclusion of educational aspects in consultations can be fundamental, since by educating patients about the importance of PE and creating a motivating environment, it is possible to increase adherence and compliance with medical recommendations, making the prescription of PE an essential component of clinical practice [25].

In addition to these challenges, time constraints during consultations and the lack of resources or accessible PE programs were also considered barriers to this practice by some specialist doctors (17.4% and 18.4%, respectively). These results are corroborated by studies carried out in Ireland [15], Mexico [35], Portugal [9], and Thailand [4], in which the lack of time for consultations is mentioned as one of the main barriers to PE prescription. In studies carried out in Saudi Arabia [18,27], Brazil [29], Canada [33], Mexico [35], Portugal [9], and Thailand [4], the scarcity of services or resources to which doctors can refer patients who need a structured PE plan is widely discussed, indicating this factor as a potential discouragement to PE prescription by doctors. The importance given to these challenges is reinforced in the systematic reviews by Hall et al. in 2022 [25] and Courish et al. in 2024 [2].

To overcome these challenges and barriers, it is crucial to adopt effective strategies such as targeted counseling, accessible goals, and greater integration of PE prescription into therapeutic plans [25]. Given the impact of PE on the prevention and treatment of chronic diseases, recognized by doctors, raising medical and public awareness, and including PE in the curriculum, institutional policies, and multidisciplinary teams can strengthen its prescription and adherence by patients [8,20].

This study has some limitations that must be recognized. First, while the study successfully gathered insights into physicians’ perceptions of the importance of PE and their counseling practices, it did not include direct feedback from patients. This absence of patient perspectives may limit the understanding of practical barriers and facilitators faced by patients, potentially overlooking key factors that influence adherence to prescribed exercise regimens. Additionally, the reliance on self-reported data from physicians may introduce self-reporting bias, as participants might overestimate their proficiency or engagement in PE counseling. Furthermore, while the findings provide valuable insights into the context of PE prescription in Portugal, the broad applicability of the results beyond this setting remains uncertain; variations in healthcare systems, cultural attitudes toward exercise, and training in PE prescription could significantly affect outcomes in different countries. Lastly, a cross-sectional design restricts the ability to capture changes in attitudes or practices over time. To address these limitations, future longitudinal research is recommended to assess changes in PE prescription practices over time, particularly in response to training programs. Such studies could measure the direct outcomes of interventions to overcome identified barriers, thus providing a more dynamic understanding of the relationship between physician PE prescriptions and patient outcomes.

It is also necessary to mention that clinicians may lack financial incentives to prioritize PE discussions with their patients, while the Portuguese healthcare reimbursement system presents financial constraints related to PE prescription, primarily compensating for direct medical care. While medications and surgeries are generally covered, supervised PE programs often incur out-of-pocket costs, limiting access for many patients. Consequently, patients with low socioeconomic status, who are often at a higher risk for chronic diseases, can find themselves at a disadvantage.

Currently, community health units and municipalities provide free or low-cost physical education programs to mitigate these barriers and attain the objectives of the Portuguese government, which are to decrease sedentary behavior and use PE as medicine. Yet adaptations to the reimbursement system are necessary to encourage PE prescriptions and enhance access for all patients.

Despite these limitations, this study has several notable strengths that enhance its contributions to the field. One of this study’s main strengths is the robustness of the sample, which includes a significant representation of the various medical specialties. This diversity not only allows the results to be generalized but also highlights the relevance of research in different clinical contexts.

The collaboration of the Centre Regional Section of the Portuguese Medical Association was crucial for the wide dissemination of the questionnaire, resulting in a substantial response rate. In addition, the elaboration of a rigorously validated questionnaire, with Cronbach’s alpha coefficients above 0.7, justifies the reliability and trustworthiness of the data collection instrument [36], showing that the questionnaire accurately captures information on PE prescription by the doctors surveyed.

In addition, it is important to emphasize that, to our knowledge, studies in this area in Portugal are scarce, making this work particularly valuable for understanding the prescription of PE by specialist doctors and for training and promoting this practice [8,9].

## 5. Conclusions

The results of this study indicate that most specialist doctors in Portugal recognize the importance of prescribing PE as a vital component in health promotion and the prevention and treatment of various pathologies. This evidence reflects a solid understanding of the benefits of PE and establishes a greater need to integrate the practice of prescribing PE into clinical consultations. However, despite this awareness, this understanding does not always translate into consistent practice in consultations, indicating an inadequate or incomplete prescription of PE. This indicates a greater need for training and knowledge of prescription guidelines.

Important barriers to the effective prescription of PE include limited adherence on the part of patients, the time available for medical consultations, and the lack of more accessible PE resources or programs.

## Figures and Tables

**Figure 1 healthcare-13-00986-f001:**
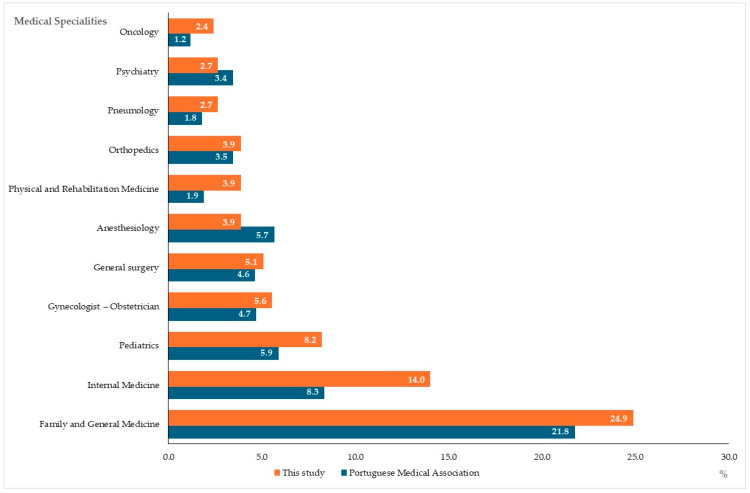
Relative distribution of responses by specialty with at least 10 responses (%), considering the proportions of specialist doctors registered in the Portuguese Medical Association and those who responded to the questionnaire. The lack of significative differences indicates the alignment between the respondents’ specialties and the registered physicians’ specialties.

**Figure 2 healthcare-13-00986-f002:**
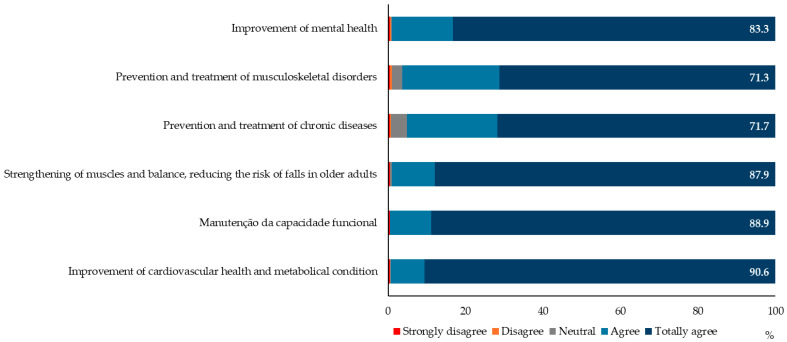
The importance of physical exercise in healthcare and health status (the percentages of the most represented class—totally agree—are shown).

**Table 1 healthcare-13-00986-t001:** Sociodemographic, academic, and professional characterization of the sample.

		n	%
** *Age classes* **			
	[29–40]	163	39.4
	]40–60]	125	30.2
	]60–83]	126	30.4
** *Sex* **			
	Female	260	62.8
	Male	154	37.2
** *End the year of Bachelor/Master classes* **			
	[1967–1975]	25	6.1
	]1975–1985]	74	18.0
	]1985–1995]	57	13.9
	]1995–2005]	62	15.1
	]2005–2018]	193	47.0
** *Medical Schools* **			
	Faculdade de Medicina da Universidade de Coimbra	227	55.2
	Faculdade de Medicina da Universidade do Porto	59	14.4
	Instituto de Ciências Biomédicas Abel Salazar-Universidade do Porto	23	5.6
	Escola de Medicina da Universidade do Minho	3	0.7
	Faculdade de Ciências da Saúde da Universidade da Beira Interior	19	4.6
	Faculdade de Medicina da Universidade de Lisboa	40	9.7
	Faculdade de Ciências Médicas|NOVA Medical School—Universidade Nova de Lisboa	26	6.3
	Foreigner Institutions	14	3.4

n—number of respondents.

**Table 2 healthcare-13-00986-t002:** Influence of age and sex of specialist doctors on intentions/habits in prescribing PE, level of competency/difficulties in this practice, and main procedures adopted (percentage in brackets).

	Age Classes				Sex		
	[29–40]	]40–60]	]60–83]	*Prob.*	Female	Male	*Prob.*
** *Choose the sentence with which you most closely identify about the prescription of physical exercise.* **							
I intend to promote and prescribe in the next few months.	23 (18.7)	6 (5.2)	4 (3.4)	<0.001	25 (11.0)	8 (6.3)	0.344
I’ve been updating and attending training courses to promote and prescribe.	9 (7.3)	6 (5.2)	3 (2.6)		10 (4.4)	8 (6.3)	
I’ve been prescribing and promoting for the last 6 months.	8 (6.5)	3 (2.6)	1 (0.9)		9 (4.0)	3 (2.4)	
I have already kept the routine of promoting and prescribing for more than 6 months now.	83 (67.5)	100 (87.0)	108 (93.1)		183 (80.6)	108 (85.0)	
** *Have you had specific/objective training in physical exercise prescription?* **							
No	89 (72.4)	93 (80.9)	87 (75.0)	0.293	181 (79.7)	88 (69.3)	0.037
Yes	34 (27.6)	22 (19.1)	29 (25.0)		46 (20.3)	39 (30.7)	
** *Are you aware of the latest physical exercise prescription guidelines (ACSM and CDC and Law no. 8932/2017 of October 10)?* **							
No	92 (74.8)	88 (76.5)	81 (69.8)	0.485	174 (76.7)	87 (68.5)	0.103
Yes	31 (25.2)	27 (23.5)	35 (30.2)		53 (23.3)	40 (31.5)	
** *Do you feel comfortable assessing the patient’s needs when prescribing physical exercise?* **							
No	68 (55.3)	60 (52.2)	24 (20.7)	<0.001	116 (51.1)	36 (28.3)	<0.001
Yes	55 (44.7)	55 (47.8)	92 (79.3)		111 (48.9)	91 (71.7)	
** *Do you feel comfortable sharing exercise recommendations with patients during consultations?* **							
No	34 (27.6)	34 (29.6)	16 (13.8)	0.009	68 (30.0)	16 (12.6)	<0.001
Yes	89 (72.4)	81 (70.4)	100 (86.2)		159 (70.0)	111 (87.4)	
** *Do you feel comfortable monitoring and adjusting physical exercise prescriptions in different cases as necessary?* **							
No	90 (73.2)	74 (64.3)	49 (42.2)	<0.001	159 (70.0)	54 (42.5)	<0.001
Yes	33 (26.8)	41 (35.7)	67 (57.8)		68 (30.0)	73 (57.5)	
** *Are you familiar with the functionality dedicated to assessment, brief counseling, and physical exercise prescription in Electronic Medical Prescription?* **							
No	59 (48.0)	84 (73.0)	74 (63.8)	<0.001	135 (59.5)	82 (64.6)	0.365
Yes	64 (52.0)	31 (27.0)	42 (36.2)		92 (40.5)	45 (35.4)	
** *When prescribing physical exercise, what is the main procedure you take during the consultation?* **							
Detailed prescription of a physical exercise plan.	4 (3.3)	4 (3.5)	6 (5.2)	0.142	4 (1.8)	10 (7.9)	0.032
Oral counseling for autonomous or supervised practice.	86 (69.9)	77 (67.0)	65 (56.0)		153 (67.4)	75 (59.1)	
Referring the patient to a supervised physical exercise program under the responsibility of a qualified professional.	19 (15.4)	23 (20.0)	36 (31.0)		49 (21.6)	29 (22.8)	
Distribution of information documents, including diverse material (audiovisual, apps, etc.).	6 (4.9)	7 (6.1)	3 (2.6)		8 (3.5)	8 (6.3)	
Use of the functionality dedicated to the assessment, brief counseling, and prescription of physical exercise in Electronic Medical Prescription.	8 (6.5)	4 (3.5)	6 (5.2)		13 (5.7)	5 (3.9)	

**Table 3 healthcare-13-00986-t003:** Distribution of specialist doctors in relation to the main challenge when prescribing or the reason for not prescribing, according to whether or not physical exercise is prescribed (percentage in brackets).

Challenge for Prescribing/Reason for Not Prescribing Physical Exercise	Physical Exercise Promotion		
	No	Yes	Total
Lack of adherence by patients	3 (5.0)	172 (48.6)	175 (42.3)
Lack of time to discuss exercises during consultations	8 (13.3)	64 (18.1)	72 (17.4)
Lack of resources or accessible exercise programs	10 (16.7)	66 (18.6)	76 (18.4)
Lack of knowledge of physical exercise	10 (16.7)	32 (9.0)	42 (10.1)
Concerns about patients’ injuries or safety	0 (0.0)	20 (5.6)	20 (4.8)
Does not apply to usual clinical practice	29 (48.3)	0 (0.0)	29 (7.0)
Total	60 (14.5)	354 (85.5)	414

## Data Availability

The data presented in this study are available on request from the corresponding author.

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
