# Peer review of "Counseling and Prescription of Physical Exercise in Medical Consultations in Portugal: The Clinician’s Perspective"

_healthcare, 2025, doi:10.3390/healthcare13090986_

Round 1

Reviewer 1 Report

Comments and Suggestions for Authors
  1. For table 2, the author needs to totally describe the reasons and explanation in the result and discussion.
  2. Methodological Rigor: The cross-sectional observational design employing a validated questionnaire is a robust approach for collecting data. The high response rate of 414 specialist physicians enhances the reliability of the findings.
  3. Statistical Analysis: The use of descriptive and inferential statistics, including appropriate tests to analyze the data, adds to the strength of the study's conclusions. The application of Cronbach's alpha demonstrates the questionnaire's reliability.
  4. Insights from Findings: The identification of barriers to PE prescription, such as patient compliance and lack of resources, provides valuable insights that can inform future interventions and training programs.
  5. Discussion Depth: The discussion section contextualizes findings within existing literature, offering a comprehensive overview of challenges and factors influencing PE prescription among physicians.
  6. Areas for Improvement
  7. Clarification of Objectives: The objective of the study could be more clearly articulated in the introduction. While the aims are mentioned, a precise statement outlining what the study intends to contribute to the existing body of knowledge would enhance reader understanding.
  8. Training and Guidelines Awareness: The manuscript notes that only 24% of physicians have received specific training on PE prescription. This raises questions about the integration of such important curricula in medical education. A more detailed discussion on how medical schools can incorporate PE training would strengthen the implications of the study.
  9. Patient-Centric Focus: While the focus on physician perceptions is vital, the study could benefit from including patient perspectives regarding PE prescriptions and adherence. Integrating qualitative data from patients could enrich the discussion on barriers.
  10. Limitations Discussion: Although limitations are mentioned, a more thorough exploration of potential biases (e.g., self-reporting bias) and the generalizability of results beyond Portugal would be beneficial.
  11. Longitudinal Study Suggestion: The study could propose future longitudinal research to evaluate changes over time in PE prescription practices and the impact of training programs. This could help measure the direct outcomes of interventions to overcome identified barriers.
  12. Formatting and Citations: Ensure that all references are consistent in format. Some citations within the text lack proper alignment with the reference section. Additionally, consider using reference management software for better organization and adherence to citation style guidelines.

Author Response

Thank you very much for taking the time to review this manuscript. Please find the detailed responses in the attachment.

Reviewer 2 Report

Comments and Suggestions for Authors

Dear Authors,

The topic of your manuscript on “Counseling and prescription of physical exercise in medical consultations: the clinician's perspective” is of interest, notably in a period when more and more people are not having a good health state and they lack physical activity due to working long hours. Nevertheless, there are some aspects that should be addressed: 

  1. When you refer to “specialist physician” what specialties you included?
  2. It is known, however, that a medical speciality would more often prescribe PE than a surgical one, such as Maxillofacial Surgery, for example. I believe that the specialities you included are not eloquent to highlight the aims of your manuscript.
  3. For example, neuroradiology, radiology, clinical pharmacology physicians are not really interacting with patients, and prescribing PE is not really within their area of expertise, this can be a reason they chose not to respond. You can add some information about this in the discussion section.
  4. Why didn`t you focus on Physical medicine and rehabilitation (PRM), Sports, Rheumatology physicians, Orthopedic surgeons and other specialities which are more prone to focus on a PE program that including all the specialities knowing, however, that this is not part of their usual approach?
  5. Also, I would exclude pediatrics from the analysis, the pediatric group should be analyzed separately.
  6. I suggest to add in the introduction section more information regarding which specialities are usually prescribing PE and why. For example, a PRM physician or Sports physician are more trained to prescribe a set of PE, while a Psychiatrist physician`s focus in not usually the PE. Please add some information in the discussion section as well.  
  7. Are PE programs supported and these patients could benefit from them in a specific environment free of charge? This is also something worth to discuss.
  8. Are these physicians paid when prescribing sets of PE? Not all specialist are familiar with the latest PE and not within their area of expertise to adapt the program, as shown by their responses as well.
  9. Is the national health system including PA and PE within the prescriptions and people can benefit without paying extra?

Good luck!

Author Response

(The authors gave the same response as above.)

Reviewer 3 Report

Comments and Suggestions for Authors

Dear authors and editor,

The manuscript entitled “Counseling and prescription of physical exercise in medical consultations: the clinician's perspective” has been evaluated. The work is scientifically important; however, adjustments need to be made to the text so that it can be submitted for publication.

I kindly ask that changes in the manuscript be highlighted in a different color so that the evaluator can find them more accurately.

Title

1- I think this title could be changed, since the only professionals qualified and authorized to prescribe physical exercises are Physical Education professionals. Therefore, I suggest that the title avoids using the word “prescription” if it is not used exclusively to talk about this professional. I believe that the word prescription could be replaced by “recommendation”, since the American College of Sports Medicine and the World Health Organization have these as principles.

Abstract

2- In line 17, please make it clear that the only professional authorized to prescribe exercise is a Physical Education professional. Other professionals can only recommend and suggest the practice following the recommendations mentioned above.

3- Adjust the objective of the abstract and the end of the introduction so that they are aligned. Remember what has been said so far: only Physical Education professionals can prescribe physical exercises. Therefore, I ask that the manuscript be revised to include this adjusted information to avoid ethical issues between the professions.

4- Between lines 22 and 25, present more details of the methodology, for example, the main questions raised, how many times the emails were sent, the period of the research and the statistical tests used.

5- In line 27, I did not understand what the statistical information “(R=0.964; p<0.001)” means. What are you analyzing to obtain this result? Make it clearer.

6- In the description of the results, also present the values ​​of “n” along with the percentages.

7- The conclusion of the abstract is very vague. Please focus on answering the objectives.

Keywords

 8- Avoid repeating words already mentioned in the title. Please replace them with synonyms.

Introduction

 9- At the beginning of the introduction, try to provide more quantitative epidemiological information regarding the world, Europe and Portugal. Present figures and percentages of mortality and morbidity caused by sedentary lifestyles and sedentary behavior. Talk about metabolic syndrome and provide current recommendations from the WHO and other entities for regular physical activity.

10- In the introduction, it was found that there are many short paragraphs that could be merged to maintain a standard size between paragraphs.

11- Another thing, it is important to reform the role of the Physical Education professional, as they are the only profession authorized to prescribe exercises; other health professionals can only recommend them according to the guidelines. This needs to be very clear so as not to affect professional ethics. This means that doctors cannot prescribe exercises, just as Physical Education professionals cannot perform surgeries or prescribe medications, for example.

Materials and Methods

12- Was the questionnaire validated internally only? With the responses of only 15 doctors?

13- Was the final version of the questionnaire sent only once? How did you access the doctors' emails? How many doctors were invited? How did you know if the professional's email was correct/updated in the system? Did you try another way to contact the professionals to encourage them to fill out the form, for example, via messaging app or social media? Please make this information clearer in the text.

14- Furthermore, since it was a questionnaire prepared by you and is not disseminated on the internet because it has not been publicly validated, please present the questions that were asked in the article or provide the full questionnaire as supplementary material, since readers will need to have access to the questionnaires in case they want to replicate it in other locations and/or times.

15- The description of the statistical analyses needs to be reorganized according to the chronology of the events. First, the data must be prepared in spreadsheets and, before being analyzed, must undergo a normalization test. Then, you will present descriptive statistics such as mean and standard deviation for normal data and median and interquartile range for nonparametric data. Then, talk about the comparison, correlation and association tests that will be used and explain how.

16- In methodology, it is also important to include all the variables that will be evaluated, such as age, height, body mass, BMI calculation, among others.

17- Please provide a study flowchart so that readers can understand the dynamics of the work.

Results

18- In line 207, you can standardize with only 1 each decimal after the comma: “49.95±14.92”

19- Present in the text the information on the average age and standard deviation of the participants' age all together and separated by sex.

20- Would you have information on the participants' body mass, height and BMI, as well as their level of physical activity and sedentary behavior? This information is important since I wonder if people care about recommending physical exercise if they do not do physical activity in their leisure time.

21- Figure 2 is not very clear where the statistical differences presented in the text are. Please adjust the figure to indicate where the differences are.

22- Figure 3 needs to present the percentages of the other answers, I suggest you describe them in the text.

23- It was not clear in the manuscript what this exercise prescription would be. Did you ask if they follow any specific guidelines regarding this possible guidance? Do they investigate the level of physical activity of patients before talking about physical exercise? All these issues are major limitations of the study. You studied prescription with a professional who is not in the exercise field and when you did the questionnaire you did not specify what the prescription would be. You did not go into the scope of intensity, duration and frequency. When we talk about prescription, we must monitor people's health parameters and in this study this issue is very vague.

Discussion

 24- The discussion talks a lot about prescription, but it doesn't talk about what exercise is and the parameters that should be followed for its prescription. The means of evaluating the components of physical fitness related to health were also not considered. I believe that all of these issues should be addressed in your study. First, you should ask your doctors if they know what physical exercise is. Then, if they understand periodization and how to put together an exercise program and its objectives.

25- Did you understand that doctors do not have adequate training to prescribe, since they probably don't know what a prescription would be? That's why I suggest changing the nomenclature to indication or recommendation. That way, even if they are laymen, they could base themselves on the American College of Sports Medicine or the World Health Organization and say that patients should follow these recommendations. However, for more appropriate monitoring, it is necessary to seek a Physical Education professional who is the person who is an expert in how to properly prescribe training.

Conclusions

 26- Remove the limitations of the study from the conclusion and move to the end of the discussion. I believe that you could reformulate these limitations, since it is not the doctor's role to prescribe. Address the research format as a limitation, since questionnaires already generate variations in responses, and when they are administered remotely, the chances of errors increase even more. Also report the lack of sociodemographic information on the participants.

Author Response

(The authors gave the same response as above.)

Round 2

Reviewer 1 Report

Comments and Suggestions for Authors

The author responded to my comments.

Author Response

Reviewer Comment

The author responded to my comments.

Authors' Answer

Dear Reviewer,

Thank you for your feedback on our manuscript. We appreciate your suggestions, as they have indeed enhanced the clarity of our work.

Best regards,

Reviewer 2 Report

Comments and Suggestions for Authors

Dear Authors,

Thank you for considering and incorporating the recommendations in your manuscript and I hope it brought more clarity to you as well as to the readers.

Good luck!

Author Response

Dear Authors,

Thank you for considering and incorporating the recommendations in your manuscript and I hope it brought more clarity to you as well as to the readers.

Good luck!

Authors' Answer

Dear Reviewer,

Thank you for your thoughtful feedback on our manuscript. We appreciate your suggestions, as they have indeed brought greater clarity to our work.

Best regards,

Reviewer 3 Report

Comments and Suggestions for Authors

Dear authors,

Thank you for providing the revised version of the manuscript. I have noticed that in Portugal it is common for doctors to prescribe physical exercises. However, it should be clear that not all countries have doctors who are able to prescribe exercises, as is the case in Brazil. Therefore, I kindly ask that you make it clear in the study methodology that this is a permitted practice in Portugal.

Author Response

Reviewer Comment

Thank you for providing the revised version of the manuscript. I have noticed that in Portugal it is common for doctors to prescribe physical exercises. However, it should be clear that not all countries have doctors who are able to prescribe exercises, as is the case in Brazil. Therefore, I kindly ask that you make it clear in the study methodology that this is a permitted practice in Portugal.

Authors answer

Dear Reviewer,

Thank you for your feedback on our manuscript. We value your insights about the variations in prescribing physical exercise across different countries. In response to your comment, we have now clarified in the methodology section (line 124) that only physicians in Portugal are authorized to prescribe physical exercise.

It is important to note that in Portugal, exercise prescriptions are viewed similarly to any other prescription. Consequently, only licensed physicians can provide customized, written prescriptions for physical exercise as part of treatment plans. The act of prescribing exercise is a medical procedure that may only be performed by authorized medical doctors.

Thank you once again for your suggestion

Best regards,